# DIFFNORM: Self-Supervised Normalization for Non-autoregressive Speech-to-speech Translation

**Weiting Tan   Jingyu Zhang   Lingfeng Shen   Daniel Khashabi   Philipp Koehn**
Department of Computer Science
Johns Hopkins University
{wtan12, jzhan237, lshen30, danielk, phi}@jhu.edu

## Abstract

Non-autoregressive Transformers (NATs) are recently applied in direct speech-to-speech translation systems, which convert speech across different languages without intermediate text data. Although NATs generate high-quality outputs and offer faster inference than autoregressive models, they tend to produce incoherent and repetitive results due to complex data distribution (e.g., acoustic and linguistic variations in speech). In this work, we introduce DIFFNORM, a diffusion-based normalization strategy that simplifies data distributions for training NAT models. After training with a self-supervised noise estimation objective, DIFFNORM constructs normalized target data by denoising synthetically corrupted speech features. Additionally, we propose to regularize NATs with classifier-free guidance, improving model robustness and translation quality by randomly dropping out source information during training. Our strategies result in a notable improvement of about $+7$ ASR-BLEU for English-Spanish (En-Es) and $+2$ ASR-BLEU for English-French (En-Fr) translations on the CVSS benchmark, while attaining over $14\times$ speedup for En-Es and $5\times$ speedup for En-Fr translations compared to autoregressive baselines.[1]

## 1  Introduction

Speech-to-speech translation (S2ST) systems are essential to bridge communication gaps and have wide application potential. We focus on non-autoregressive modeling for direct *speech-to-speech* translation, converting source speech to the target without intermediate text data. Such direct S2ST systems [25, 23, 22, 32, 31, 21, 20] can preserve non-linguistic information, avoid error propagation from cascaded systems [29, 34] (e.g., a combination of speech recognition and machine translation systems), and achieve faster inference speed.

Non-autoregressive Transformers (NAT) [21, 20] has played a central role in current S2ST work [31, 32, 21]. NAT translates source waveforms into target **speech units** via parallel decoding, achieving performance comparable to or better than autoregressive models while greatly reducing inference time. This process is often referred to as *speech-to-unit* (S2UT) translation [31]. The predicted speech units are then converted to target waveforms via a unit vocoder in the *unit-to-speech* synthesis stage [32, 38]. However, NATs suffer from incoherent and repetitive generations, referred to as the **multi-modality problem** [12]. This issue stems from NAT's assumption of conditional independence during parallel decoding, worsened by the complex and multi-modal nature of training data distribution.

In this work, we propose DIFFNORM, a **self-supervised** speech normalization strategy that alleviates the multi-modality problem of NAT models by simplifying the target distribution. Instead of distilling training data from an autoregressive model [10] or utilizing perturbed speech to train a normalizer [32, 21], we rely on the denoising objective of Denoising Diffusion Probabilistic Models [15, DDPM] to normalize target speech units. DIFFNORM inject synthetic noise to speech features and use diffusion

---

[1] Code available at: https://github.com/steventan0110/DiffNorm

38th Conference on Neural Information Processing Systems (NeurIPS 2024).

model to gradually recover the feature, obtaining a simplified and more consistent data distribution that obscures non-crucial details. As the denoising objective is learned in a self-supervised manner over latent speech representations, it eliminates the necessity for transcription data [32] or manually crafted perturbation functions [21]. As illustrated in Fig. 1, applying DIFFNORM to the target data obtains normalized speech units that lead to better NAT training for speech-to-unit translation.

Besides using DIFFNORM as a data-centric strategy to mitigate the multi-modality problem, we also propose a regularization strategy to enhance the NAT model's robustness and generalizability when facing complex data distribution (e.g., linguistic diversity and acoustic variation [21, 20]). Inspired by classifier-free guidance [17, CG], during training, we occasionally drop out source information and replace it with a "null" representation, compelling the models to generate coherent units without conditioning on the source data. During iterative parallel decoding of the NAT model, we obtain higher-quality translation by mixing conditional and unconditional generation. Ultimately, combining DIFFNORM and CG results in our top-performing state-of-the-art system, achieving approximately $+7$ and $+2$ ASR-BLEU increment for En-Es and En-Fr translation compared to previous non-autoregressive systems on the CVSS [24] benchmark.

In conclusion, we alleviate the multi-modality problem by proposing (1) diffusion-based normalization and (2) regularization with classifier-free guidance. To the best of our knowledge, we are the first to adapt diffusion models and classifier-free guidance into speech-to-speech translation and NAT modeling. Our methods obtain notable improvement compared to previous systems and maintain fast inference speed inherent in non-autoregressive modeling, achieving speedups of $14\times$ for En-Es and $5\times$ for En-Fr compared to autoregressive baselines.

## 2 Problem formulation and overview

We aim to develop a direct (textless) speech-to-speech translation system that transduces a source speech $\boldsymbol{x} = (x_1, \cdots, x_N)$ into target speech. We follow Lee et al. [31] to reduce speech-to-speech translation into two sub-tasks: speech-to-unit translation and unit-to-speech synthesis. In this work, we focus on speech-to-unit translation and follow prior work [31, 32, 21] to use the same unit-to-speech component for a fair comparison. To generate speech units for the target language, we first extract speech feature $\boldsymbol{h} = (h_1, \cdots, h_M) \in \mathbb{R}^{M \times H}$, where each feature has $H$ dimensions. Subsequently, a K-means clustering model is trained on extracted features and used to generate speech units $\boldsymbol{y} =$

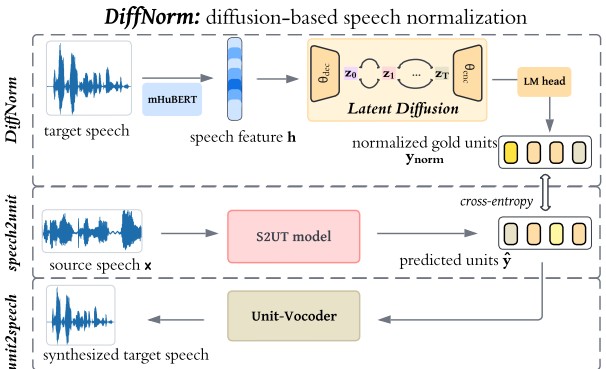

Figure 1: Overview of our proposed system. We first normalize the target speech units with the denoising process from the latent diffusion model. Then speech-to-unit (S2UT) model is trained to predict normalized units, which are converted into waveform from an off-the-shelf unit-vocoder.

$(y_1, \cdots, y_M) \in \mathbb{R}^{M \times 1}$. Once the source speech $\boldsymbol{x}$ and target speech unit $\boldsymbol{y}$ are prepared, we train sequence-to-sequence models to translate from source speech into target units. In practice, we follow Lee et al. [32] to use a multilingual-HuBERT (mHuBERT) model to extract $H = 768$ dimensional speech features $\boldsymbol{h}$. Then we use a 1000-cluster K-means model to predict speech units given the encoded features.[2] For speech-to-unit translation, we follow Huang et al. [21] to adopt Conditional Masked Language Modeling [10, CMLM], a kind of non-autoregressive transformer (NAT).

To mitigate NAT models' multi-modality problem, we propose DIFFNORM (section §3), which denoises synthetically corrupted speech features to construct normalized speech units $\boldsymbol{y}_{\text{norm}}$. As shown in Fig. 1, such normalized units are then used to train the S2UT (CMLM) model, which generates better-translated units for the unit vocoder to synthesize target speech.

---

[2] We take the mHuBERT and K-means models off-the-shelf to produce units consistent with prior work. See github.com/facebookresearch/fairseq/tree/main/examples/speech_to_speech

Additionally, we propose to incorporate classifier-free guidance [17, CG], a widely adopted strategy for diffusion-based image generation, as a regularization method to improve the non-autoregressive speech-to-unit system (section §4). As shown in Fig. 3, by forcing NAT models to unmask target speech units without conditioning on source information, the model becomes more robust, generating coherent speech units that result in higher-quality translations. During inference, we follow classifier-free guidance to mix conditional and unconditional generation for the NAT model's iterative decoding, further enhancing its translation quality.

## 3 DIFFNORM: denoising diffusion models for speech normalization

Previous normalization strategy [32, 21] on speech-to-unit translation relies on connectionist temporal classification (CTC) fine-tuning [4] using the HuBERT [19] model. For example, Lee et al. [32] rely on single-speaker data from VoxPopuli [58] to produce normalized (speaker-invariant) speech units for HuBERT-CTC fine-tuning. On the other hand, Huang et al. [21] generate acoustic-agnostic units by perturbing rhythm, pitch, and energy information with (manually) pre-defined transformation functions. We propose to normalize speech units using self-supervised denoising objectives from Denoising Diffusion Probabilistic Models [15, DDPM]. We believe DDPM is more suitable for speech normalization because: (1) It only requires monolingual speech data, without the need of transcriptions to create a text-to-unit model as in [32]. (2) It learns to denoise features in a high-dimensional space rather than relying on hand-designed perturbation as in [21]. DIFFNORM consists of a variational auto-encoder (VAE) and a diffusion model. Since vanilla VAE [27] and DDPM [15] are applied to image generation, we modify them to support token generation and incorporate multitasking objectives. The VAE model reduces the dimension of the speech feature, mapping feature $h$ into lower dimensional latent $z$. The diffusion model (visualized in Fig. 2) is then trained to denoise on the latent representation space, aiming to recover the original latent $z_0$ from the standard Gaussian noise $z_T$.[3] In the subsequent sections, we begin by outlining the architecture of our VAE model (§3.1). Then, we delve into how the diffusion model is trained using the latent variables encoded by the VAE (§3.2).

### 3.1 Variational auto-encoders for latent speech representation

Since speech features encoded by mHuBERT have a high dimension ($H = 768$), we compress the feature into lower-dimension latents for the diffusion model. The VAE model consists of an encoder, a decoder, and a language modeling head. Following our problem formulation (§2), we first prepare a sequence of target speech features $h = (h_1, \cdots, h_M) \in \mathbb{R}^{M \times H}$ and their corresponding speech units $y = (y_1, \cdots, y_M)$. Our VAE model's encoder will map the feature into lower dimension latent $z = f(h; \theta_{\text{enc}}) \in \mathbb{R}^{M \times Z}$ where $Z < H$ is the pre-defined latent dimension. Then VAE model's decoder reconstructs the speech feature $\hat{h} = f(z; \theta_{\text{dec}})$ and we apply the language modeling head to convert the reconstructed feature into a distribution over speech units' vocabulary $v = f(\hat{h}; \theta_{\text{lm}}) \in \mathbb{R}^{M \times V}$. Following prior work [21, 32], we cluster speech features into $V = 1000$ units. The training objective is a weighted combination of reconstruction loss ($\mathcal{L}_{\text{recon}}$), negative log-likelihood (NLL) loss ($\mathcal{L}_{\text{nll}}$), and a Kullback–Leibler (KL) divergence term resulted from the Gaussian constraint [27] to regularize the representation space:

$$\mathcal{L} = \lambda_1 \mathcal{L}_{\text{recon}} + \lambda_2 \mathcal{L}_{\text{nll}} + \lambda_3 \mathcal{L}_{\text{kl}}, \tag{1}$$

where the reconstruction loss is defined as $\mathcal{L}_{\text{recon}}(\hat{h}, h) = ||h - \hat{h}||^2$, and the NLL loss is computed as the cross-entropy between ground-truth units $y$ and the predicted vocabulary distribution $v$: $\mathcal{L}_{\text{nll}}(v, y) = -\sum_{i=1}^{M} y_i \log v_i$, where $y_i$ is the one-hot version of $y_i$. Lastly, $\mathcal{L}_{\text{kl}}$ can be solved analytically when we assume Gaussian distribution for both prior and posterior approximation of latent $z$ (more details in [27] and Appendix B). Though the Gaussian constraint has a small weight $\lambda_3$, we show in the ablation study (§6.2) that regularizing latent distribution is critical for the diffusion model. For details on the architecture of our VAE model, we direct readers to Appendix B.

### 3.2 Diffusion model for denoising latent speech representation

**Training**   Once the VAE model is trained, we encode speech feature as the first step feature $z_0 = f(h; \theta_{\text{enc}})$ for the diffusion model. Diffusion models consists of a (1) forward process that

---

[3] The subscript $z_0$ is added to indicate the time step for diffusion process and $z_0$ and $z$ are equivalent.

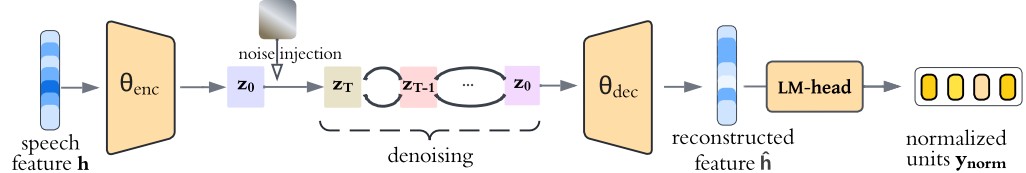

Figure 2: Visualization of our latent diffusion model's denoising process for speech normalization. The clean latent $z_0$ is synthetically noised (into $z_T$) and the reverse diffusion process gradually denoise it to generate normalized speech units.

gradually transforms $z_0$ into a standard Gaussian distribution, and a (2) reverse process that denoise and recovers the original feature $z_0$. Following DDPM [15], with a pre-defined noise scheduler (let $\beta_t \in (0,1)$ be the scaling of noise variance, define $\alpha_t = 1 - \beta_t$ and denote $\bar{\alpha}_t = \prod_{s=1}^{t} \alpha_t$ as the noise level for time $t$), the forward and reverse process can be written as:

$$q(z_t|z_0) = \mathcal{N}(z_t; \sqrt{\bar{\alpha}_t}z_0, (1-\bar{\alpha}_t)\epsilon) \quad p_\theta(z_{t-1}|z_t) = \mathcal{N}(z_{t-1}; \mu_\theta(z_t,t), \sigma^2\mathbf{I}) \tag{2}$$

where mean $\mu_\theta(z_t, t)$ and variance $\sigma^2$ are parameterized by trainable models, typically based on U-Net [44] or Transformer [57]. We follow DDPM to train models using the re-weighted noise estimation objective:

$$\mathcal{L}_{\text{noise}}(\theta, t) = \mathbb{E}_{x_0,\epsilon,t}||\epsilon - \epsilon_\theta(z_0, t)||^2 \tag{3}$$

where the network learns to predict the injected Gaussian noise $\epsilon$ from the forward process. Besides noise estimation, we also train the model with auxiliary reconstruction and NLL loss using VAE model's decoder and language modeling head. Our training procedure is summarized in Alg. 1.

---

**Algorithm 1** Latent Diffusion Model Training

1: **Input**: speech feature $h$, speech units $y$.
2: **Pre-compute**: $\beta_t, \alpha_t, \bar{\alpha}_t, t \in [1, T]$
3: **while** not converged :
4:    $z_0 = f(h; \theta_{\text{enc}})$
5:    $t \sim \mathcal{U}[1, T], \epsilon \sim \mathcal{N}(0, \mathbf{I})$
6:    $z_t = \sqrt{\bar{\alpha}_t}z_0 + \sqrt{1 - \bar{\alpha}_t}\epsilon$
7:    $\hat{\epsilon} = \epsilon_\theta(z_t, t)$
8:    $\mathcal{L}_{\text{noise}} = ||\epsilon - \hat{\epsilon}||^2$
9:    $\hat{z}_0 = (z_t - \sqrt{1 - \bar{\alpha}_t}\hat{\epsilon})/\sqrt{\bar{\alpha}_t}$
10:   $\hat{h} = f(\hat{z}_0; \theta_{\text{dec}}), \mathcal{L}_{\text{recon}} = ||h - \hat{h}||^2$
11:   $v = f(\hat{h}; \theta_{\text{lm}}), \mathcal{L}_{\text{nll}} = \text{NLL}(v, y)$
12:   Take gradient descent step with loss
13:      $\mathcal{L} = \gamma_1 \mathcal{L}_{\text{noise}} + \gamma_2 \mathcal{L}_{\text{recon}} + \gamma_3 \mathcal{L}_{\text{nll}}$

**Algorithm 2** Normalized Units Construction

1: **Input**: speech feature $h$, start time $T$
2: **Pre-compute**: $\beta_t, \alpha_t, \bar{\alpha}_t, t \in [1, T]$
3: $z_0 = f(h; \theta_{\text{enc}})$
4: $z_T = \sqrt{\bar{\alpha}_T}z_0 + \sqrt{1 - \bar{\alpha}_T}\epsilon$
5: $t = T$
6: **while** $t > 0$ :
7:    $\hat{\epsilon} = \epsilon_\theta(z_t, t)$
8:    $\hat{z}_0 = (z_t - \sqrt{1 - \bar{\alpha}_t}\hat{\epsilon})/\sqrt{\bar{\alpha}_t}$
9:    $z_{t-1} = \sqrt{\bar{\alpha}_{t-1}}\hat{z}_0 + \sqrt{1 - \bar{\alpha}_{t-1}} \cdot \hat{\epsilon}$
10:   $t = t - 1$
11: $\hat{h} = f(z_0; \theta_{\text{dec}})$
12: $y_{\text{norm}} = \text{argmax} f(\hat{h}; \theta_{\text{lm}})$
13: **return** denoised units $y_{\text{norm}}$

---

As shown in Alg. 1, we randomly sample the current timestep $t \in [1, T]$ and compute corresponding scheduling parameters $\beta_t, \alpha_t, \bar{\alpha}_t$[4]. The training process involves the regular DDPM objective that injects Gaussian noise for the model $\epsilon_\theta$ to perform noise estimation (Alg. 1, line 6-8). Additionally, we generate the pseudo latent $\hat{z}_0$ by reversing the noise injection process (line 9), which is then decoded by the VAE into speech features and units for reconstruction loss (line 10) and NLL loss (line 11). Finally, the objective is a weighted sum of noise estimation loss, reconstruction loss, and NLL loss:

$$\mathcal{L} = \gamma_1 \mathcal{L}_{\text{noise}} + \gamma_2 \mathcal{L}_{\text{recon}} + \gamma_3 \mathcal{L}_{\text{nll}} \tag{4}$$

In our analysis (§6.2), we show that adding NLL and reconstruction loss is indeed helpful in improving the diffusion model's reconstruction quality. Lastly, to parameterize our diffusion model for noise estimation, we modify Diffusion Transformer [37, DiT] to suit our task. For details of our architecture and hyper-parameters, we refer readers to Appendix B.

**Inference** For speech normalization, we choose a start time $T \in [0, 200]$ that decides the amount of noise to inject for the diffusion model to recover. Given the start time $T$, we follow Denoising Diffusion Implicit Models [48, DDIM] sampler to reverse noised latent $z_T$ back to $z_0$. Then, our VAE model converts $z_0$ back to speech units with its decoder and language modeling head. Our inference procedure is summarized in Alg. 2 and visualized in Fig. 2.

---

[4] We follow the cosine schedule proposed by Nichol and Dhariwal [35] with a maximum timestep of $T = 200$

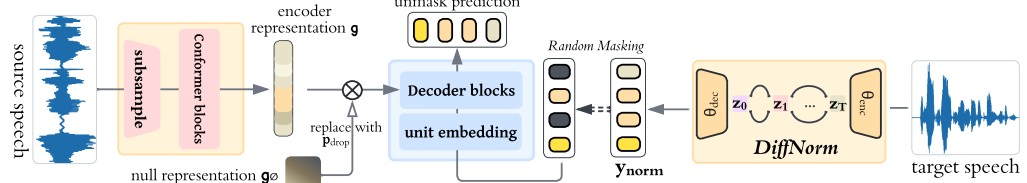

Figure 3: Visualization of CMLM for speech-to-unit translation where the model is trained with the unmasking objective to recover $\boldsymbol{y}_{\text{norm}}$. When classifier-free guidance is used, with probability $p_{\text{drop}}$, we replace the encoded source speech $\boldsymbol{g}$ by a "null" representation $\boldsymbol{g}_\emptyset$.

Note that since our diffusion model is used to normalize speech units as a data preprocessing strategy, inference speed is not a concern. Therefore we use step size $\Delta t = 1$ for DDIM sampling throughout our experiments. However, the inference speed could be easily improved by setting a larger step size.

## 4    Classifier-free guidance for non-autoregressive transformer

In §3, we proposed DIFFNORM to normalize speech units and obtained speech-unit pairs $(\boldsymbol{x}, \boldsymbol{y}_{\text{norm}})$ that benefit NAT training. In this section, we propose to adapt classifier-free guidance [17] to regularize NAT models (visualized in Fig. 3), further enhancing NAT-based S2UT model's translation quality.

**Training**    We largely adhere to standard CMLM training [21] but introduce a small dropout probability $p_{\text{drop}} = 0.15$ for the encoder representations. Specifically, we parameterize the encoder and decoder of the CMLM as $\phi_{\text{enc}}$ and $\phi_{\text{dec}}$, respectively. The encoder processes the source speech into a representation $\boldsymbol{g} = f(\boldsymbol{x}; \phi_{\text{enc}})$. The decoder then predicts the vocabulary distribution $\boldsymbol{v} = f(\hat{\boldsymbol{y}}|\boldsymbol{g}; \phi_{\text{dec}})$ from $\boldsymbol{g}$ and the randomly masked target speech units $\hat{\boldsymbol{y}}$. The total amount of masked token is uniformed sampled: $n \sim \mathcal{U}[1, M]$; subsequently, $n$ of $M$ tokens from the target units $\boldsymbol{y}$ are randomly masked to form noisy target units $\hat{\boldsymbol{y}}$. CMLM also trains a length predictor that estimates the output length given input sequences and we refer readers to [21] for more details.

Using the predicted distribution $\boldsymbol{v}$, we compute the NLL loss against the ground truth units $\boldsymbol{y}$ at the masked positions to train the model. If dropout is applied, the decoder receives a "null" representation $\boldsymbol{g}_\emptyset$—a randomly-initialized learnable vector—forcing it to rely solely on the target information to unmask units. For more details, we refer readers to Appendix C and previous paper [10, 21].

**Inference**    CMLM generates tokens through iterative unmasking. Given the input sequence $\boldsymbol{x}$, CMLM first predicts the length of output sequence $M$ and initialize a sequence of $M$ masked tokens $\hat{\boldsymbol{y}}_0 = ([\text{mask}]_1, \cdots, [\text{mask}]_M)$. Given the total number of iterations $T$[5] and current iteration $t \in [1, T]$, CMLM decodes all tokens $y_i, i \in [1, M]$ in parallel using their probabilities:

$$y_i^t = \underset{w}{\arg\max} \, \mathbb{P}(y_i = w | \boldsymbol{x}, \hat{\boldsymbol{y}}_{t-1}; \phi_{\text{dec}}) \quad p_i^t = \log \mathbb{P}(y_i^t | \boldsymbol{x}, \hat{\boldsymbol{y}}_{t-1}; \phi_{\text{dec}}) \tag{5}$$

From the generated tokens $\hat{\boldsymbol{y}}_t = (y_1^t, \cdots, y_M^t)$, we replace $k = M \cdot \frac{T-t}{T}$ tokens with the lowest probabilities $p_i^t$ with mask tokens. This re-masked sequence $\hat{\boldsymbol{y}}_t$ is then inputted into the CMLM for further decoding iterations. This process continues until the final step $t = T$, at which point no tokens are re-masked. With our proposed regularization from classifier-free guidance, we compute two vocabulary distribution in each iteration step $t$:

$$p_{\text{orig}}^t = \mathbb{P}(\cdot | \boldsymbol{g}, \hat{\boldsymbol{y}}_{t-1}; \phi_{\text{dec}}) \quad p_{\text{uncond}}^t = \mathbb{P}(\cdot | \boldsymbol{g}_\emptyset, \hat{\boldsymbol{y}}_{t-1}; \phi_{\text{dec}}) \tag{6}$$

where $p_{\text{orig}}^t$ is the same as vanilla CMLM and $p_{\text{uncond}}^t$ is the unconditional probability obtained with the "null" representation. Then we select and re-mask tokens using the adjusted probability

$$p^t = \omega(p_{\text{orig}}^t - p_{\text{uncond}}^t) + p_{\text{orig}}^t \tag{7}$$

where the hyper-parameter $\omega$ is used to control the degree to push away from the unconditional distribution. In the special case of $\omega = 0$, only conditional distribution is used during iterative decoding, resulting in the same generation process as standard CMLM. Through our intensive analysis

---

[5] $T$ was used in §3 to denote total timesteps for diffusion models. We abuse the notation and re-use it here to represent the number of decoding iterations for CMLM.

(Appendix F), we determine that a relatively small $\omega$ (e.g., $\omega = 0.5$) gives the best result, especially when the number of decoding iterations is large (i.e., $T > 10$). Notably, even when $\omega = 0$ — which corresponds to the traditional CMLM decoding method — CMLMs regularized with classifier-free guidance during training can consistently outperform their standard counterparts. This observation further validates the effectiveness of our proposed regularization.

## 5 Experiments

### 5.1 Experimental setup

**Dataset** We perform experiments using the established CVSS-C datasets [24], which are created from COVOST2 by employing advanced *text-to-speech* models to synthesize translation texts into speech [59]. CVSS-C comprises aligned speech in multiple languages along with their respective transcriptions. Our methods are evaluated on two language pairs: English-Spanish (En-Es) and English-French (En-Fr), with detailed data statistics provided in Table 1. As our focus is on direct

| Split | En-Es | | En-Fr | |
|-------|-------|--------|-------|--------|
| | Size | Length | Size | Length |
| Train | 79,012 | 256 | 207,364 | 228 |
| Valid | 13,212 | 296 | 14,759 | 264 |
| Test | 13,216 | 308 | 14,759 | 283 |

Table 1: Data statistics for CVSS benchmarks. Length is the average number of speech units of the target speech.

speech-to-speech translation, transcriptions are solely utilized for evaluation purposes. Utilizing the speech data from CVSS, we preprocess the target speech to generate speech units using the mHuBERT and K-means model, as described in our problem formulation (§2).

**Evaluation** To evaluate the performance of various speech-to-speech translation systems, we adopt the standard methodology established in previous studies [32, 31, 21] to compute the ASR-BLEU score. Firstly, our speech-to-unit translation system generates speech units based on the input speech, which are then transformed into speech waveforms using a unit-vocoder. The unit-vocoder is built upon the HifiGAN architecture [28] with customized objectives, detailed in Appendix D. Once we have the waveforms, we employ an ASR model to transcribe them and calculate the BLEU score against the reference transcriptions. This ASR model is fine-tuned based on the WAV2VEC2.0 [5] model with the CTC objective. We direct readers to the original paper [7] for more details. Both the unit-vocoder and ASR models are sourced from an off-the-shelf repository, ensuring consistency in evaluation methodology with previous studies [32, 21].[6]

**Normalized dataset construction** We adhere to the procedure outlined in Alg. 2 to generate normalized speech units. By manipulating start time $T$, we control the level of noise in the latent $z_T = \sqrt{\bar{\alpha}_t} z_0 + \sqrt{1 - \bar{\alpha}_t} \epsilon$ for the diffusion model, balancing between the reconstruction quality and normalization effect. We explore different levels of noise injection and choose $T = 100$ for En-Es and $T = 120$ for En-Fr (further details in §6.1). Hence, we adopt these settings to construct our normalized dataset, CVSS-NORM.

**Compared systems** In this section, we provide brief descriptions of evaluated speech-to-unit models. For autoregressive models, we evaluate the Transformer model trained following Lee et al. [31]. We also assess the Conformer model trained similar to the Transformer, but with its encoder replaced by a Conformer-Encoder [14]. Lastly, the Norm Transformer shares the same architecture as the Transformer, but it is trained on normalized speech units that are speaker-invariant. The normalized dataset is constructed following the strategy proposed by Lee et al. [32].

For non-autoregressive systems, we train the Conditional Masked Language Model (CMLM) following Huang et al. [21], using a Conformer-based encoder and a Transformer decoder. The CMLM + Bilateral Perturbation (BiP) system retains the same architecture as CMLM but is trained on normalized speech units constructed with BiP [21].

For our improved systems, we train CMLM + DIFFNORM, which shares the same architecture as CMLM but is trained on CVSS-NORM, the normalized dataset obtained through DIFFNORM. Additionally, we train the CMLM + CG model that incorporates classifier-free guidance introduced in §4. Finally, the CMLM + DIFFNORM + CG system uses the architecture of CMLM + CG and is trained on our normalized dataset CVSS-NORM.

---

[6] Repository available at: github.com/facebookresearch/fairseq/examples/speech_to_speech/asr_bleu.

## 5.2 Results

Table 2 summarizes the (ASR-BLEU) performances of various S2UT systems. Note that, for non-autoregressive methods, Table 2 shows their results obtained with 15 decoding iterations. For more details on the inference speedup achieved through non-autoregressive modeling, we plot Fig. 4 to provide details on the "quality vs. latency" trade-off. Observations from Table 2 are:

**DIFFNORM greatly enhances translation quality compared to systems using original speech units** (model 6 vs. 4; 8 vs. 7). For example, CMLM with DIFFNORM achieves about $+7$ BLEU score on En-Es compared to CMLM trained on original units. **DIFFNORM also outperforms previous normalization strategies** (model 6 v.s. 2, 5), validating the superior quality of normalized speech units obtained through our methods. Besides normalization, classifier-free guidance effectively improves speech-to-unit quality as a regularization strategy, leading to better translation quality (model 7 v.s. 4; 8 v.s. 6). Finally, **combining both classifier-free guidance and DIFFNORM (model 8) results in the best overall system, outperforming both autoregressive and non-autoregressive baselines.**

| ID | System | Quality ↑ | | | Inference Speed ↑ | |
|---|---|---|---|---|---|---|
| | | En-Es | En-Fr | Fr-En* | Speed | Speedup |
| **Autoregressive** | | | | | | |
| 1 | Transformer[†] [31] | 10.07 | 15.28 | 15.44 | 870 | 1.00× |
| 2 | Norm Transformer[†] [32] | 12.98 | 15.93 | 15.81 | 870 | 1.00× |
| 3 | Conformer[†] | 13.75 | 17.07 | 18.02 | 895 | 1.02× |
| **Non-autoregressive Model** | | | | | | |
| 4 | CMLM | 12.58 | 15.62 | 16.95 | 4651 | 5.34× |
| 5 | CMLM + BiP[†][21] | 12.62 | 16.97 | 18.03 | | |
| **Our Improved Non-autoregressive Model** | | | | | | |
| 6 | CMLM + DIFFNORM | 18.96 | 17.27 | **19.53** | 4651 | 5.34× |
| 7 | CMLM + CG[‡] | 17.06 | 16.89 | _ | | |
| 8 | CMLM + DIFFNORM + CG[‡] | **19.49** | **17.54** | _ | | |

Table 2: Comparison of speech-to-speech models evaluated by quality (ASR-BLEU) and speed (units/seconds).*: Fr-En experiments are added during author response period and we leave model 7,8 for future work. Results with [†] are taken from the prior work [21]. [‡] We use $w = 0.5$ for CG. **Our NAT models achieve superior translation quality while maintaining their fast inference speed**.

**Decoding speed**    In Fig. 4, we illustrate the "quality-latency" trade-off of various non-autoregressive speech-to-unit systems. Quality is measured using ASR-BLEU, while latency is determined by the relative speedup over the autoregressive system, calculated as "generated units/second". For instance, the first marker on the line plot represents 15 decoding iterations, resulting in a speedup of 5.34× compared to the autoregressive baseline. Additionally, we include the performance of the Conformer-based autoregressive model as a horizontal dashed line. We observe that our improved system consistently outperforms the baseline CMLM model [21] and achieves better performance than the autoregressive model with more than 14× speedup for En-Es and 5× speedup for En-Fr.

## 6 Analysis

### 6.1 Effect of synthetic noise injection

In this section, we explore the effects of varying degrees of noise injection on DIFFNORM. Recall (from Alg. 2) that synthetic noise is injected into the latent representation as $z_T = \sqrt{\bar{\alpha}_T}z_0 + \sqrt{1 - \bar{\alpha}_T}\epsilon$. Hence, adjusting the start time $T$ allows us to control the degree of noise injection.

**Setup and Evaluation**    We perturb the degree of noise injection by varying $T$, and assess the *reconstruction quality* of normalized speech units and *downstream performance* of CMLM models trained with different normalized units. For reconstruction quality, we measure the accuracy (**Acc-Rec**) of reconstructed units and ASR-BLEU (**BL-Rec**) of synthesized speech when using original

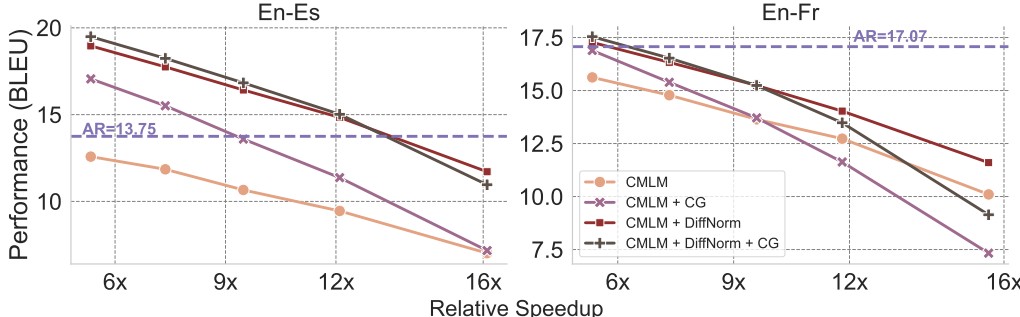

Figure 4: Trade-off between quality (ASR-BLEU) and latency for varying numbers of decoding iterations. Five markers correspond to {15, 10, 7, 5, 3} decoding iterations. Decreasing the number of iterations results in a decline in model performance, traded off for faster speedup. With DIFFNORM and CG, **our S2UT model achieves a better quality-latency trade-off** than CMLM and outperforms a strong autoregressive baseline with large speedups.

units/speech as the reference. For downstream performance, we report the downstream ASR-BLEU of translated speech produced by CMLM models trained with the particular noise setup (**BL-Dn**).[7]

**Results**   Shown in Table 5, increasing $T$ leads to more noise and a decline in reconstruction quality, which aligns with expectations that higher noise levels pose greater challenges for the diffusion model in restoring original features accurately. Interestingly, we find from Table 3 that when $T$ is too small or large (e.g., $T = 50$ or $T = 150$), the normalized units do not result in an ideal downstream system. We hypothesize that there is barely any normalization effect when $T$ is too small; and when $T$ is too large, the reconstructed units are no longer semantically correct. This phenomenon is visualized in Fig. 5, where we plot the log-mel spectrograms of the reconstructed speech. As more noise is introduced (larger $T$), the reconstructed speech becomes smoother (e.g., portions of blank speech are filled in by diffusion), exhibiting a more pronounced deviation from the original speech.

| *Start Step* | *Scheduler Parameters* | | | *En-Es* | | | *En-Fr* | | |
|---|---|---|---|---|---|---|---|---|---|
| | $\beta_t$ | $\sqrt{\bar{\alpha}_t}$ | $\sqrt{1-\bar{\alpha}_t}$ | **Acc-Rec** ↑ | **BL-Rec** ↑ | **BL-Dn** ↑ | **Acc-Rec** ↑ | **BL-Rec** ↑ | **BL-Dn** ↑ |
| original units | N/A | N/A | N/A | 100 | 48.7 | 12.58 | 100 | 40.64 | 15.62 |
| $T = 50$ | 0.007 | 0.917 | 0.398 | **89.4** | **48.5** | 18.56 | **91.1** | **40.38** | 17.12 |
| $T = 100$ | 0.016 | 0.697 | 0.717 | 81.2 | 47.63 | **18.96** | 83.0 | 39.9 | 16.69 |
| $T = 120$ | 0.022 | 0.577 | 0.816 | 75.3 | 46.25 | 16.7 | 77.7 | 39.03 | **17.27** |
| $T = 150$ | 0.038 | 0.373 | 0.928 | 57.8 | 31.43 | 14.77 | 60.0 | 28.3 | 14.03 |

Table 3: For different start steps $T$, we show corresponding noise scheduling parameter values, reconstruction quality (**-Rec** columns), and downstream translation quality (**-Dn** column). **Noise injection that perturbs about 20% of units (i.e., 80% Acc-Rec) results in the best downstream S2UT performance (highlighted in bold text).**

From the Table 3, we observed a substantial enhancement in En-Es translation performance with $T = 50$. Consequently, we conducted additional experiments for En-Es translation with start times $T = 10$ and $T = 30$, detailed in Table 4. The results show that at a minimal start time of $T = 10$, the reconstruction accuracy reaches 93.8%, yielding a downstream ASR-BLEU score of 15.98. This

| *Start Step* | Acc-Rec | BL-Rec |
|---|---|---|
| $T = 10$ | 93.8 | 15.98 |
| $T = 30$ | 91.6 | 17.92 |

Table 4: Reconstruction and downstream performance with small noise injection.

score significantly surpasses the baseline CMLM result of 12.58. The high accuracy indicates that about 6% of tokens vary post-reconstruction, likely due to the VAE model's regularization effect since the diffusion model does not cause notable reconstruction deviations at such a small $T$. This suggests that the Spanish speech dataset might have considerable acoustic variations and noise, which the VAE model can partially mitigate. As $T$ increases, the diffusion model further refines the representation, leading to improved downstream results, as evidenced by the rise in ASR-BLEU score from from

---

[7]ASR-BLEU is computed by synthesizing speech from the reconstructed units and transcribing it using the ASR model, as detailed in §5.1.

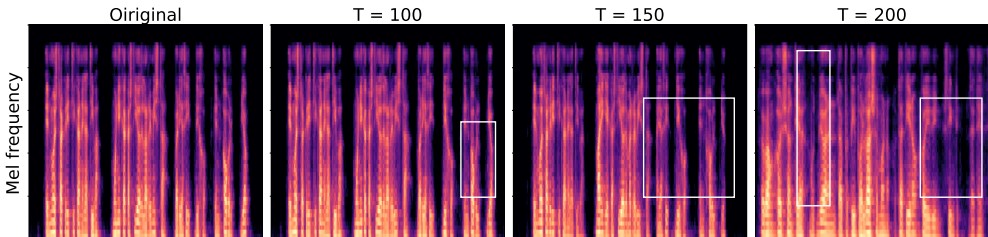

Figure 5: Visualization of reconstructed speech's log-mel spectrograms. Noticeable divergence from the original speech is highlighted in the white bounding boxes.

$T = 10$ to $T = 100$. This ablation study underscores that **the optimal choice of T is affected by the dataset quality**.

## 6.2 Ablation on training objectives for DIFFNORM

DIFFNORM requires an auto-encoder to map speech features into lower-dimension latents and then train denoising diffusion models using the encoded latents. In this section, we perform ablation experiments to investigate the effect of (1) latent dimension (2) Gaussian constraints (3) multitask objective for diffusion models. Firstly, to investigate the effect of latent dimension, we train auto-encoders that encode speech features into 16, 32, and 128 dimensions. As shown in Table 5, it comes as no surprise that a larger latent dimension yields superior reconstruction results, evidenced by higher accuracy across systems trained with varying objectives. Next, we turn our attention to the importance of applying Gaussian constraints to the auto-encoder's latent representation space. Our results (comparing with and without **KL**) reveal that incorporating Gaussian constraints is crucial. Latent spaces not

| Start Step | Objectives | | Latent Dimension | | |
|---|---|---|---|---|---|
| | KL | Multitask | 16 | 32 | 128 |
| $T = 50$ | ✗ | ✓ | 51.4 | 69.1 | 85.8 |
| | ✓ | ✗ | 80.9 | 86.4 | 89.3 |
| | ✓ | ✓ | **80.9** | **86.5** | **89.4** |
| $T = 100$ | ✗ | ✓ | 13.3 | 32.5 | 73.1 |
| | ✓ | ✗ | 64.5 | 76 | 80.8 |
| | ✓ | ✓ | **65.3** | **76.2** | **81.2** |
| $T = 150$ | ✗ | ✓ | 2.9 | 5.3 | 34.6 |
| | ✓ | ✗ | 19.8 | 38.2 | 56.8 |
| | ✓ | ✓ | **20.2** | **40** | **57.8** |

Table 5: Accuray of reconstructed speech units. **KL**: when applied, the latent space is regularized to be Gaussian [27]. **Multitask**: when not applied, the latent diffusion model is trained only with $\mathcal{L}_{\text{noise}}$.

regularized by $\mathcal{L}_{\text{kl}}$ lead to significantly poorer performance. Finally, we explore the effectiveness of our proposed multitasking objective for the diffusion model by training another vanilla DDPM solely with $\mathcal{L}_{\text{noise}}$ (no **Multitask**). We find that the diffusion model's reconstruction capability indeed improves when employing the multitasking objective, particularly when denoising from a more noisy latent representation (e.g., $T = 150$). Through our ablation analysis, we identify the optimal setup for our DiffNorm model, which involves (1) mapping speech features to 128 dimensions, (2) regularizing latent space by Gaussian constraints, and (3) utilizing the multitask objective for diffusion training.

## 7 Related work

**Direct speech-to-speech translation (S2ST)** Direct S2ST aims to directly translate speech into another language without cascaded systems that rely on transcriptions or translations. Translatotron [25] and Translatotron 2 [23] are among the first systems for direct S2ST, which uses sequence-to-sequence model to map source speech into spectrograms. Then, a spectrogram decoder is used to synthesize the target language's speech. Instead of transducing speech into spectrogram, Tjandra et al. [53], Zhang et al. [62] utilize Vector-Quantized Variational Auto-Encoder (VQ-VAE) [56] to discretize target speech and convert *speech-to-speech* translation into a *speech-to-unit* task. More recently, Lee et al. [31] improved the speech-to-unit models by obtaining such units with a k-means clustering model trained on self-supervised representation from HuBERT [19]. To convert units back to speech, Lee et al. [31] follows Polyak et al. [38] to train a unit-vocoder based on HifiGAN [2].

**Speech normalization** Speech representation are typically extracted from pre-trained encoders [19, 3, 40, 5] and can be compressed or adapted in different speech-to-speech/text tasks [61, 63, 50, 52, 51]. Inspired by previous work on speech enhancement [54, 1], Lee et al. [32] propose to

normalize speech by synthesizing speaker-invariant waveforms through text-to-speech (TTS) systems [60, 46, 42, 41]. Huang et al. [21] propose to normalize speech with Bilateral Perturbation that focus on the rhythm, pitch, and energy information. Different from previous normalization methods that requires transcription [32] or manually designed perturbation [21], our DIFFNORM strategy leverages diffusion models. **Diffusion models** [15, 48, 49] have achieved remarkable generative ability to produce high-quality images [8, 16, 43, 37, 45] and audio [33, 39, 47]. Using self-supervised denoising objectives [15], we train effective denoising models capable of normalizing speech for training better speech-to-unit systems.

**Non-autoregressive speech-to-speech translation** For sequence-to-sequence modeling, autoregressive [18, 6, 57] and non-autoregressive [12, 13, 10, 20] models have been widely explored. Lee et al. [31] reduce speech-to-speech into speech-to-unit task and follow the widely-used modeling strategy, Transformer [57], to predict speech units. Later, non-autoregressive models [21, 9] have been explored and Huang et al. [21] is the first to use a non-autoregressive transformer model, Conditional Masked Language Model [10, CMLM], for speech-to-unit task. Fang et al. [9], on the other hand, adopts Directed Acyclic Transformer [20] for speech-to-unit translation.

# 8    Conclusion

We improve speech-to-unit translation system through (1) corpus distillation by constructing normalized speech units with DIFFNORM and (2) regularization with classifier-free guidance. Our improved non-autoregressive (NAR) models, greatly outperform previous NAR models, yielding an increase of approximately $+7$ and $+2$ BLEU points for En-Es and En-Fr translation, respectively. Notably, DIFFNORM and classifier-free guidance maintain the inference speed advantages inherent in NAR models. Consequently, our approach obtains better performance to autoregressive baselines while achieving over $14\times$ speedup for En-Es and $5\times$ speedup for En-Fr translations.

## Acknowledgement

We express our profound appreciation to anonymous reviewers for their helpful suggestions. We also thank Xiuyu Li for his valuable suggestions, which greatly enriched our work. Lastly, we thank JHU + Amazon Initiative for Interactive AI for sponsoring the work.

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

# Supplementary Material

## A Qualitative Examples

From §5.2 we demonstrate the effectiveness of DiffNorm and classifier-free guidance by evaluating ASR-BLEU scores. In this section, we provide example transcriptions of generated Spanish and French speech in Table 6 and Table 7. Compared to the vanilla CMLM model that has erroneous or incomplete translation, our method results in higher-quality translation.

| Model | Generated Translation | English Version |
|---|---|---|
| Reference | este trabajo fue posible en cooperación con otros como alphonse milne-edwards y leon vaillant. | This work was made possible in cooperation with others such as Alphonse Milne-Edwards and leon vaillant |
| CMLM | esta obra fue posible para lugar con otros como alphon y leol vigelante | This work was possible to place with others such as Alphon and Leol Vigelante |
| CMLM + CG | este hora fue hizo por la cooperación con otros con walphonxe mill y leon fallent | This time was made by cooperation with others with Walphonxe Mill and Leon Fallent |
| CMLM + DiffNorm | esta obra fue posible la cooperación con otros como al von milo edwad y leon val-lente | This work was possible through cooperation with others such as Al von Milo Edwad and Leon valente |
| CMLM + DiffNorm + CG | este trabajo fue posible por la cooperación con otros como alphonso edos y león va-lente | This work was possible through the co-operation with others such as Alphonso Edos and León Valente |

Table 6: Example transcription of translated Spanish speech from different systems.

## B Diffusion Model Hyperparameters

### B.1 Variational Auto-Encoder

**Architecture**: Our Auto-encoder consists of an encoder, a decoder, and a language modeling head. The encoder architecture follows WaveNet [55] that combines Convolutional Neural Networks [30] with residual connections. In practice, we use 2 stacks of WaveNet Residual Blocks, where each block has 3 layers. Our decoder uses the same architecture as the encoder to revert the latent dimension back

| Model | Generated Translation | English Version |
|---|---|---|
| Reference | mais la principauté est envahie et le prince-évêque joseph-clément de bavière doit s'exiler | but the principality is invaded and the prince-bishop Joseph-Clement of Bavaria must go into exile |
| CMLM | mais la précicipauté et le prince joseph clémont de bavier est fisé en exil | but the precicipality and prince joseph clemont of bavier is exiled |
| CMLM + CG | mais la principauté est lebséve josefh crémend de bavière et contré en exil | but the principality is lebséve josefh cremend of bavaria and countered in exile |
| CMLM + DiffNorm | la princepôuté est envahie et le prince-évêque joser clémant de bavière est contr | the princepôuté is invaded and the prince-bishop joser clemant of bavaria is opposed |
| CMLM + DiffNorm + CG | mais la principauté est envahie et le prend s évêque joseph de bavier et borgé en exile | but the principality is invaded and takes bishop joseph of bavaria and borgé into exile |

Table 7: Example transcription of translated French speech from different systems.

to the original dimension. Additionally, we apply a Transformer [57] encoder on top of the decoded feature to further enhance model capacity. The Transformer-encoder has the same configuration as the vanilla Transformer-base model (6 layers, 8 attention heads, etc.). Lastly, to decode speech units from the feature, we use a language modeling head which is parameterized by a feedforward network that converts feature dimension ($H = 768$) into vocabulary size ($V = 1000$).

**Training**: To train the VAE model, as described in §3.1, we use a combination of three objectives

$$\mathcal{L} = \lambda_1 \mathcal{L}_{\text{recon}} + \lambda_2 \mathcal{L}_{\text{nll}} + \lambda_3 \mathcal{L}_{\text{kl}} \tag{8}$$

where $\lambda_1 = 100, \lambda_2 = 1, \lambda_3 = 0.001$. Now we provide more details about $\mathcal{L}_{\text{kl}}$: in practice, we follow [27] to model the latent $\boldsymbol{z}$ by estimating its mean $\boldsymbol{\mu}$ and variance $\boldsymbol{\sigma}$ using the encoder. With the re-parameterization trick, the latent is sampled as $\boldsymbol{z} = \boldsymbol{\mu} + \boldsymbol{\sigma} \cdot \boldsymbol{\epsilon}$ where $\boldsymbol{\epsilon} \sim \mathcal{N}(0, \mathbf{I})$. By constraining the mean and variance to follow a Gaussian prior, Kingma and Welling [27] showed that

$$\mathcal{L}_{\text{kl}} = \frac{1}{2} \sum_{j=1}^{J} (1 + \log((\sigma_j)^2) - (\mu_j)^2 - (\sigma_j)^2) \tag{9}$$

We implement our VAE model on Fairseq [36]. For optimization, we use the Adam [26] optimizer with betas $(0.9, 0.98)$ and we apply gradient clipping by setting –clip-norm=2.0. During training, we apply dropout with a probability of 0.1. We train the VAE model using a learning rate of 5e-4 with distributed data-parallel (DDP) on 4 A100 GPUs, where we set the maximum batch token to be 15000.

### B.2 Latent Diffusion Model

**Architecture**: We modified DiT [37] to design a Transformer-based architecture for noise estimation. Since our input latent is a sequence of encoded speech feature $\boldsymbol{z} \in \mathbb{R}^{M \times Z}$, we first apply a 1D Convolution to convert the latent to our model's hidden dimension (set to 512). Then we also encode the timestep with a learnable Sinusoidal Positional Embedding [57] to obtain the time embedding.

Subsequently, we feed the latent and time embedding to our modified WaveNet Block, which applies an affine transformation of the latent using the time embedding, similar to the "Scale and Shift" operation in DiT's adaptive layer norm (adaLN) block. For our diffusion model, we use 4 stacks of such modified WaveNet Block, each containing 8 layers. The output feature from WaveNet is then passed through a Transformer-Encoder (12 layers with 8 attention heads) to further enhance model capacity. Lastly, a projection layer parameterized by the feedforward network is used to compress the transformed feature to the latent dimension, which is then used for noise estimation in equation (3).

**Training**: As described in §3.2, we train the diffusion model with a multitask objective:

$$\mathcal{L} = \gamma_1 \mathcal{L}_{\text{noise}} + \gamma_2 \mathcal{L}_{\text{recon}} + \gamma_3 \mathcal{L}_{\text{nll}} \tag{10}$$

and we empirically select $\gamma_1 = 1, \gamma_2 = 0.25, \gamma_3 = 0.005$. For optimization, our latent diffusion model has the same setting as our VAE model (with Adam optimizer, clip-norm equals 2.0, and dropout with 0.1 probability). We train the diffusion model using a learning rate of 1e-4 with DDP on 4 A100 GPUs, where we set the maximum batch token to be 12000. The model is warmup-ed by 10000 steps.

# C  Non-autoregressive Transformer Details

## C.1  Background on Non-Autoregressive Transformers

Non-autoregressive Transformers [12, 13, 10, *inter alia.*] is a family of models that transduce sequences. In this work, we follow the formalization from the widely-used NAT: Conditional Masked Language Model [10, CMLM]. CMLM consists of an encoder $\phi_{\text{enc}}$ that represents the input as high-dimensional features and a decoder $\phi_{\text{dec}}$ that generates tokens conditioning on the encoded features. Different from auto-regressive Transformers [57], CMLM's decoder does not apply causal masking and is trained with an unmasking objective.

**CMLM Training** Following formulation from §2, given the input speech $\boldsymbol{x} = (x_1, \cdots, x_N)$ and target speech units $\boldsymbol{y} = (y_1, \cdots, y_M)$, CMLM first mask the target units into $\hat{\boldsymbol{y}}$. The total amount of masked token is uniformed sampled: $n \sim \mathcal{U}[1, M]$; subsequently, $n$ of $M$ tokens from the target units $\boldsymbol{y}$ are randomly masked to form noisy target units $\hat{\boldsymbol{y}}$. Then, the decoder generates a distribution over vocabulary conditioning on the encoder representation and noisy target units $\boldsymbol{v} = f(\hat{\boldsymbol{y}}|\boldsymbol{g}; \phi_{\text{dec}})$ where $\boldsymbol{g} = f(\boldsymbol{x}; \phi_{\text{enc}})$ is the encoder representation. Cross-entropy loss is then computed for the masked positions to update model parameters. CMLM also trains a length predictor that estimates the output length given input sequences and we refer readers to [21] for more details.

**CMLM Mask-Predict**: For CMLM inference, an iterative decoding scheme is used through unmasking speech units and re-masking low-confident positions. Given the input sequence $\boldsymbol{x}$, CMLM first predicts the length of output sequence $M$ and initialize a sequence of $M$ masked tokens $\hat{\boldsymbol{y}}_0 = ([\text{mask}]_1, \cdots, [\text{mask}]_M)$. Given the total number of iterations $T$ and current iteration $t \in [1, T]$, CMLM decodes tokens across all positions and also computes their log-probabilities:

$$y_i^t = \underset{w}{\arg\max}\, \mathbb{P}(y_i = w | \boldsymbol{x}, \hat{\boldsymbol{y}}_{t-1}; \phi_{\text{dec}}) \quad p_i^t = \log \mathbb{P}(y_i^t | \boldsymbol{x}, \hat{\boldsymbol{y}}_{t-1}; \phi_{\text{dec}}) \tag{11}$$

From the generated tokens $\hat{\boldsymbol{y}}_t = (y_1^t, \cdots, y_M^t)$, we replace $k = M \cdot \frac{T-t}{T}$ tokens with the lowest probabilities $p_i^t$ with mask tokens. This re-masked sequence $\hat{\boldsymbol{y}}_t$ is then inputted into the CMLM for further decoding iterations. This process continues until the final step $t = T$, at which point no tokens are re-masked.

## C.2  Architecture

We follow the same architecture and implementation from Huang et al. [21], where a Conformer-based encoder is used to obtain representation from the source speech and a Transformer-decoder (with no causal masking) is used to generate units. The Conformer-encoder has 12 layers with 512 hidden dimensions and 9 attention heads. The encoder also contains a CNN-based sub-sampler that has an effective stride size of 320 (i.e., it reduces the length of speech input by 320×). The Transformer-decoder has 6 layers with 8 attention heads and uses a fixed positional embedding for speech units. When classifier-free guidance is used, we randomly dropout encoding from the Conformer model with a probability of 15% as described in §4.

## C.3  Hyper-parameter for Model Training

We train the model with a learning rate of 5e-4, using 10000 warmup steps We apply Adam optimizer with betas $(0.9, 0.98)$ and we clip the gradient by setting –clip-norm=10. We apply dropout with 0.1 probability and we use label smoothing of ratio 0.2 when computing the NLL loss. We train the model with DDP on 4 A100 GPUs, with a maximum batch tokens of 40,000.

# D  Unit-to-Speech Synthesis

We follow prior work [32, 38, 41] to convert predicted units (from S2UT model) into speech waveforms. Specifically, given units $\boldsymbol{y}$, we use the HifiGAN-based unit-vocoder from [38] to synthesize speech. The unit-vocoder is trained using a generator $\mathcal{G}$ and discriminator network $\mathcal{D}$. During training, the generator is updated with reconstruction loss based on the mel-spectrogram of the true and synthesized waveforms. Additionally, adversarial loss and feature-matching loss are added to enhance the fidelity of generated speech. To train the discriminator, Polyak et al. [38] follows the minmax objective from

GAN [11]. After training, the discriminator is discarded and only the generator is used to synthesize waveforms.

We follow [32] to use a unit-vocoder that is also trained with a duration predictor so that it can synthesize speech with reduced speech units. For more details, we refer readers to the discussion from the original paper [32, 41], as well as the open-sourced repository: `github.com/facebookresearch/fairseq/examples/speech_to_speech`

## E  Limitations and Broader Impacts

**Limitations**: The major limitation of the work is the data pre-processing cost from DIFFNORM, as our method requires inferencing diffusion model on all training samples to obtain better speech units. With the CVSS benchmark, both En-Es and En-Fr have less than 1M pairs, which can be processed easily with our resource (4 A100 GPUs). However, when the dataset scales up, the corpus distillation cost could be large.

**Broader Impacts**: Our system helps bridge the communication gap between users of different languages. However, as our system involves speech synthesis, it also has a risk of being used to mimic a particular speaker.

## F  Ablation on Classifier-free Guidance

In this section, we compare non-autoregressive transformers trained with and without classifier-free guidance (CG). We train CMLM models with classifier-free guidance and evaluate them with 5, 10, 15 iterations of decoding. From Table 8, we observe that CG improves the quality of translation whether the model is trained on original or normalized speech units. We find CG brings more improvement on the original units than the normalized units. This happens because normalized speech units are more conformed and already result in a large improvement in their translation quality, making the regularization effect from CG less obvious. Nevertheless, the best-performing system is achieved with both DIFFNORM and CG.

Comparing different hyperparameters $w$, we find a small value like $w = 0.5$ or $w = 1$ brings the most improvement empirically, and such improvements are more noticeable when the number of decoding iterations is larger. For example, comparing the results under 5 and 15 iterations, we find $w = 0$ gives better results when the number of iterations is small while $w = 0.5$ obtains the best performance with 15 iterations.

| Model | En-Es | | | En-Fr | | |
|---|---|---|---|---|---|---|
| | 5 | 10 | 15 | 5 | 10 | 15 |
| CMLM | 9.45 | 11.85 | 12.58 | **12.73** | 14.78 | 15.62 |
| CMLM + CG (w=0) | **12.22** | **15.58** | 16.62 | 12.13 | 15.23 | 16.65 |
| CMLM + CG (w=0.5) | 11.37 | 15.51 | **17.06** | 11.63 | **15.39** | **16.89** |
| CMLM + CG (w=1) | 11.27 | 15.4 | 16.99 | 11.52 | 15.44 | 16.65 |
| CMLM + CG (w=2) | 10.98 | 14.97 | 16.57 | 11.12 | 15.04 | 16.50 |
| CMLM + CG (w=3) | 10.63 | 14.7 | 16.22 | 10.80 | 14.71 | 16.17 |
| CMLM + DiffNorm | 14.84 | 17.75 | 18.96 | **14.03** | 16.33 | 17.27 |
| CMLM + DiffNorm + CG (w=0) | 14.69 | 17.57 | 18.63 | 13.03 | 16.05 | 17.13 |
| CMLM + DiffNorm + CG (w=0.5) | **15.02** | **18.24** | **19.49** | 13.48 | **16.53** | **17.54** |
| CMLM + DiffNorm + CG (w=1) | 14.77 | 18.18 | 19.37 | 13.29 | 16.27 | 17.48 |
| CMLM + DiffNorm + CG (w=2) | 14.03 | 17.65 | 18.9 | 12.8 | 15.97 | 17.15 |
| CMLM + DiffNorm + CG (w=3) | 13.45 | 17.09 | 18.5 | 12.23 | 15.44 | 16.52 |

Table 8: Speech-to-speech translation performance of CMLM models with different CG hyperparameters.

# G  Experiment Result Tables

| Model | Number of Iterations | | | | |
|---|---|---|---|---|---|
| | 3 | 5 | 7 | 19 | 15 |
| CMLM | 7.02 | 9.45 | 10.66 | 11.85 | 12.58 |
| + CG | 7.17 | 11.37 | 13.59 | 15.51 | 17.06 |
| + DiffNorm | 11.71 | 14.84 | 16.42 | 17.75 | 18.96 |
| + DiffNorm + CG | 10.96 | 15.02 | 16.83 | 18.24 | 19.49 |

Table 9: Experimental Results of different En-Es speech-to-unit translation systems.

| Model | Number of Iterations | | | | |
|---|---|---|---|---|---|
| | 3 | 5 | 7 | 19 | 15 |
| CMLM | 10.1 | 12.73 | 13.64 | 14.78 | 15.62 |
| CMLM + CG | 7.33 | 11.63 | 13.71 | 15.39 | 16.89 |
| CMLM + DiffNorm | 11.6 | 14.03 | 15.24 | 16.33 | 17.27 |
| CMLM + DiffNorm + CG | 9.14 | 13.48 | 15.24 | 16.53 | 17.54 |

Table 10: Experimental Results of different En-Fr speech-to-unit translation systems.

