# OpenReview forum: "DiffNorm: Self-Supervised Normalization for Non-autoregressive Speech-to-speech Translation"
_NeurIPS.cc/2024/Conference — NeurIPS 2024 poster_

### Official Review · Reviewer_DRdm · 2024-07-11

**Soundness:** 2
**Presentation:** 3
**Contribution:** 2
**Rating:** 5
**Confidence:** 3

**Summary:**

This work builds on TRANSPEECH (Huang et al., 2023) by applying the diffusion method to reduce noise, thereby normalizing speech units for further generation. The authors further use classifier-free guidance to enhance non-autoregressive generation. They conduct experiments on CVSS En-Fr and En-Es datasets, comparing their methods with the baseline. Both proposed methods show improvements.

**Strengths:**

1. This work proposes a diffusion method to normalize target speech units, which outperforms previous approaches.

2. It explores classifier-free guidance to improve NAT generation for the speech-to-speech translation task.

**Weaknesses:**

1. The paper overstates its contributions. The authors claim to be the first to apply diffusion in the speech-to-speech translation task. However, they merely use it to generate auxiliary training targets and still follow the S2UT strategy (Lee et al., 2022b). This diminishes the novelty of the paper.
2. Although the work is based on TRANSPEECH, the authors compare only two of the three translation directions. One of the core contributions is applying the diffusion method to normalize speech units, yet the method shows minimal improvement (only 0.3 BLEU) compared to BiP (Bilateral Perturbation) in the En-Fr task. Given the method significantly outperforms others in the En-De task, this result is perplexing.
3. For a minor suggestion, in Tables 3 and 4, ensure consistent significant figures. Additionally, it would be more appropriate to place Table 3 before Table 4.

**Questions:**

1. How does the performance fare on the CVSS Fr-En task?
2. In my opinion, the goal of normalizing the unit is to disentangle linguistic information from noisy speech features. If you use DDPM for this process, the target $z_0$, which is the output of the VAE encoder, should be the linguistic feature. How do you ensure that $z_0$ represents the desired linguistic information?

**Limitations:**

The authors appropriately state the limitations and broader impacts of this work.

---

> ### Author Rebuttal · Authors · 2024-08-06
>
> We appreciate the time and effort you have dedicated to reviewing this paper. We are glad that you find our diffusion-based normalization and regularization strategy effective. Below are our responses to your comments:
>
> > However, they merely use it to generate auxiliary training targets and still follow the S2UT strategy. This diminishes the novelty of the paper.
> >
>
> We believe that innovative data-centric methods represent significant contributions to the field, even if the diffusion model is not directly employed as the speech-to-speech translation model. In fact, it is explored and known that the diffusion model struggles with tasks that have weak source-target dependencies, such as translation [3].
>
> Our method distinguishes itself from previous normalization techniques [1,2] by requiring only target-side information and being learned in a self-supervised manner. Through empirical studies on En-Es, En-Fr, and the recently added experiments on Fr-En, our methods have consistently demonstrated substantial improvements for non-autoregressive S2UT models. As reviewer AuR5 noted, “This technique has wide applicability. All NAT S2S unit-based models can potentially benefit from it. Not to mention it requires no handcrafting rules for noise injection.” This endorsement underscores the potential and versatility of our approach in enhancing speech translation models.
>
> > The method shows minimal improvement (only 0.3 BLEU) compared to BiP (Bilateral Perturbation) in the En-Fr task. Given the method significantly outperforms others in the En-De task, this result is perplexing.
> >
>
> The effectiveness of our normalization strategy is indeed dependent on the dataset. It is important to emphasize that our method should **primarily be compared with the CMLM baseline, rather than CMLM + BiP**, because our contribution lies in a self-supervised normalization strategy combined with a regularization strategy. These are completely orthogonal to BiP’s rule-based normalization approach. Our best system achieves an ASR-BLEU improvement of approximately 7 points on En-Es, 2 points on En-Fr, and 2.5 points on Fr-En over the CMLM baseline. In contrast, the improvement from BiP over the CMLM baseline is typically around or less than 1 point.
>
> When comparing the improvements between En-Es and En-Fr, the 7-point increase for En-Es is indeed very impressive. However, this does not diminish the significance of the 2-point improvement on En-Fr. We believe three factors contribute to the substantial gain observed in En-Es:
>
> 1. **Dataset Size**: The En-Es dataset is smaller, which means that the impact of normalization on the dataset is more pronounced, leading to more noticeable improvements.
> 2. **Speech Length**: Spanish speech is, on average, longer than French speech, as indicated in Table 1. Longer sequences could be more susceptible to acoustic variations, which our normalization method effectively mitigates.
> 3. **Background Noise**: Spanish speeches from the CommonVoice dataset also contain background noise. Our proposed normalization strategy can remove this noise, thereby enhancing the clarity and quality of the speech data and benefiting the training of NAT models.
>
> These factors combined suggest that while our method is broadly effective, its impact is particularly significant in scenarios where the data exhibits specific challenges that our approach directly addresses.
>
> > For a minor suggestion, in Tables 3 and 4, ensure consistent significant figures. Additionally, it would be more appropriate to place Table 3 before Table 4.
> >
>
> Thanks for the suggestion! We will adjust it accordingly to make it more consistent and appropriate.
>
> > How does the performance fare on the CVSS Fr-En task?
> >
>
> Thanks for the question. We kindly refer you to our response to all reviewers, where we show the result for Fr-En task as this is requested by multiple reviewers. Consistent with our paper’s results, DiffNorm also improves upon baseline CMLM by more than 2.5 ASR-BLEU on Fr-En (and more than 1.5 ASR-BLEU compared to CMLM+BiP). We hope this new set of results could further validate the effectiveness of our method and address your concerns.
>
> > How do you ensure that 𝑧0 represents the desired linguistic information?
> >
>
> Since the Variational Autoencoder (VAE) and the diffusion model are trained in a self-supervised manner on the target feature, they achieve high reconstruction quality and maintain linguistic information when properly trained. This is evidenced by the ASR-BLEU scores for the reconstructed units (BL-Rec column in Table 4), which show that transcription from reconstructed speech units achieves performance comparable to the original units (48.5 vs. 48.7 for Spanish and 40.38 vs. 40.64 for French speech). If the reconstructed units failed to preserve linguistic information, the speech synthesized from such units would exhibit significantly worse ASR-BLEU results.
>
> It is important to note that CVSS-C is not a large dataset. By expanding the dataset size, we anticipate even higher quality reconstruction and a more pronounced normalization effect that reduces acoustic variations while preserving the linguistic content of the speeches.
>
> ----
>
> References:
>
> [1] Lee et al., (2022). Textless speech-to-speech translation on real data.
>
> [2] Huang et al., (2023). TranSpeech: Speech-to-Speech Translation With Bilateral Perturbation
>
> [3] Tan, X. (2024). Lessons From the Autoregressive/Nonautoregressive Battle in Speech Synthesis.

---

> ### Author Response · Authors · 2024-08-12
> **Invitation for Comments and Clarifications**
>
> Dear Reviewer DRdm,
>
> We greatly value your feedback and have provided clarifiications to your questions and additional experiments on Fr-En task. To ensure that we have properly addressed your concerns, we would greatly appreciate if you could review our responses and provide any further comments. We are looking forward to engaging with you before the discussion period ends.
>
> Thank you for your time and consideration.

---

### Official Review · Reviewer_AuR5 · 2024-07-13

**Soundness:** 3
**Presentation:** 3
**Contribution:** 3
**Rating:** 7
**Confidence:** 4

**Summary:**

This paper proposes DiffNorm, a diffusion-based self-supervised method for speech data normalization, aiming to alleviate multimodal problem in non-autoregressive speech-to-speech translation (NAT). DiffNorm consists of a VAE to reconstruct the speech feature and a diffusion model to add and remove noise of latent vector. Experiment shows that DiffNorm significantly improve NAT translation quality compared to baselines.

**Strengths:**

1. It is surprising to see pure data normalization improves NAT so much. ASR-BLEU improves 7 BLEU on En-Es direction with DiffNorm.
2. This technique has wide applicability. All NAT S2S unit-based models can potentially benefit from it. Not to mention it requires no handcrafting rules for noise injection.
3. Ablation studies on noise level and training of DiffNorm further provide users a general guide on how to adapt DiffNorm to their own dataset and model.

**Weaknesses:**

1. The multi-modal problem has two aspects: semantic and acoustic. DiffNorm seems able to reduce acoustic modalities. Unclear if DiffNorm can also do that on semantic modalities, i.e., multiple feasible translations for the same source input.
2. Classifier-free guidance combined with DiffNorm leads to worse performance than DiffNorm alone in Figure 4 and the authors ignore it in the text. Elaboration is needed here.
3. Lack baseline comparison with several latest S2ST models like TransSentence [1], PolyVoice [2], SeamlessM4T and etc.

[1] TranSentence: speech-to-speech Translation via Language-Agnostic Sentence-Level Speech Encoding without Language-Parallel Data.
[2] PolyVoice: Language Models for Speech to Speech Translation
[3] SeamlessM4T—Massively Multilingual & Multimodal Machine Translation

**Questions:**

1. Issues listed in weakness.
2. What is the role of VAE here? Can we drop it and directly use a diffusion model here for denoising?
3. Table 3 mentions larger dimension of latent vector brings better performance, why not use the original number of dimensions instead of compressing it?
4. Why improvement on En-Fr is not that significant compared to En-Es? How does language direction impact the performance?
5. Is there way to visualize the noise added on the latent vectors? Like what does it mean for the audio?

**Limitations:**

Besides what I have already mentioned in the weakness, experiments in the paper are only conducted on En-X, but not reverse. Also, it would be interesting to see how DiffNorm works on non-European languages like Chinese and Japanese.

---

> ### Author Rebuttal · Authors · 2024-08-06
>
> We appreciate the time and effort you have dedicated to reviewing this paper. We are excited to see that you find our method effective and widely applicable! Below are our responses to your comments:
>
> > DiffNorm seems able to reduce acoustic modalities. Unclear if DiffNorm can also do that on semantic modalities
> >
>
> Like previous studies [1,2], DiffNorm also targets acoustic modalities. By injecting random Gaussian noise into all positions, DiffNorm effectively removes background noise and simplifies acoustic variations while preserving linguistic information. However, due to its design, DiffNorm is unable to simplify semantic modalities. Addressing this limitation, we plan to explore methods to simplify semantic modalities for speech units in future work, with using an auto-regressive model to generate output as a potential starting point [3,4]. Although the challenge of semantic multimodality persists, we believe that mitigating acoustic variations is crucial, and we have demonstrated significant improvements with our method.
>
> > Classifier-free guidance combined with DiffNorm leads to worse performance than DiffNorm alone in Figure 4 and the authors ignore it in the text
> >
>
> Thank you for bringing this to our attention! We will include a discussion on this issue in our revised draft. As you may observe from Figure 4 and Table 7 (in the Appendix), CG performs poorly when the number of decoding iterations is small (e.g., 5 iterations). This occurs because, during inference, the logits are slightly perturbed in each decoding iteration (refer to eq.7). In the early decoding iterations, the probability difference ($p_{orig} - p_{uncond}$) may provide poor guidance, **as $p_{uncond}$, modeled from a sequence of [MASK] tokens, is essentially random**. In later iterations, $p_{uncond}$ becomes more useful because the sequence is already partially filled. When the number of iterations is larger, only a few most confident predictions are retained, allowing the model to disregard the early iteration’s negative influence from $p_{uncond}$ . However, when the total number of decoding iterations is small, more incorrect predictions are retained, which negatively impacts future generation.
>
> > Lack baseline comparison with several latest S2ST models like TransSentence, PolyVoice, SeamlessM4T and etc
> >
>
> Thank you for bringing these notable works to our attention. We will incorporate a discussion to compare with them in our revised draft. Overall, we believe our contribution is orthogonal to these baselines.
>
> For TransSentence, they employ an auto-regressive model based on the Transformer architecture, with its major novelty being the use of language-agnostic sentence-level encoding. Since they tested on the same dataset as ours, we can directly compare our results with theirs. As indicated in their result table (Table 2), our performance is superior: En→Fr (17.54 vs. 14.69), En→Es (19.49 vs. 18.9), and Fr→En (19.53 vs. 16.59).
>
> For PolyVoice and SeamlessM4T, both utilize much larger training datasets and are based on autoregressive modeling and utilize multitask training (though SeamlessM4T incorporates NAR T2U decoding). Given these differences, a direct comparison with these systems may not be fair. We believe our contributions in speech normalization and CG regularization for non-autoregressive speech-to-speech translation are distinct from these baselines.
> Nevertheless, we appreciate your pointing out these related works, and we will ensure to add a more thorough discussion in our paper to address these comparisons.
>
> > What is the role of VAE here? Can we drop it and directly use a diffusion model here for denoising?
> >
>
> The role of the Variational Autoencoder (VAE) in our framework is twofold: (1) to reduce the feature dimension, making it more efficient for diffusion model training, and (2) to regularize the latent space with Gaussian constraints, which is crucial for the training of the diffusion model. The necessity of the VAE, particularly (2), is underscored in our ablation study. As demonstrated in Table 3, the reconstruction quality is significantly compromised when the Gaussian constraint is not applied, highlighting the essential role of the VAE in our methodology.
>
>
> > Table 3 mentions larger dimension of latent vector brings better performance, why not use the original number of dimensions instead of compressing it?
> >
>
> Yes, technically it is possible to train a Variational Autoencoder (VAE) that maps features to their original dimension, but this approach would be highly inefficient. The larger the dimension, the more computationally expensive it becomes to train and perform inference with diffusion models. Utilizing the original dimension would significantly prolong the training process and increase costs for the diffusion model, with only marginal improvements in performance. Therefore, we have limited our experiments to a maximum dimension of 128.
>
> Moreover, while a dimension of 128 provides the best reconstruction quality, using a smaller dimension for the latent space could potentially achieve similar downstream results. This can be accomplished by appropriately adjusting the T value, which controls the amount of noise injection. We plan to explore the impact of varying the latent dimension and adjusting the T value in future research to optimize performance and efficiency.

---

> ### Author Response · Authors · 2024-08-06
> **Rebuttal by Authors (Cont'd)**
>
> This comment follows from **Rebuttal by Authors**.
>
> > Why improvement on En-Fr is not that significant compared to En-Es? How does language direction impact the performance?
> >
>
> Our method has demonstrated effective performance on both language pairs, with a notable 2 BLEU point improvement for En-Fr, which is considered substantial. However, the more dramatic improvement observed in the En-Es pair can likely be attributed to specific characteristics of the dataset. Three key factors may contribute to this significant enhancement:
>
> 1. **Dataset Size**: The En-Es dataset is smaller, which means that the impact of normalization on the dataset is more pronounced, leading to more noticeable improvements.
> 2. **Speech Length**: Spanish speech is, on average, longer than French speech, as indicated in Table 1. Longer sequences could be more susceptible to acoustic multimodality, which our normalization method effectively mitigates.
> 3. **Background Noise**: The Spanish speeches from the CommonVoice dataset contain background noise. Our proposed normalization technique is capable of removing such noise, thereby enhancing the clarity and quality of the speech data.
>
>
> > Is there way to visualize the noise added on the latent vectors? Like what does it mean for the audio?
> >
>
> For visualization, please refer to Figure 5, which displays the log-mel spectrogram of the reconstructed speech. The extent of corruption in the speech varies based on the level of noise injection. As the noise injection level changes, the speech feature can be quantized into speech units that produce sounds ranging from completely unrecognizable noise to those closely resembling the original audio.
>
> > experiments in the paper are only conducted on En-X, but not reverse. Also, it would be interesting to see how DiffNorm works on non-European languages like Chinese and Japanese.
> >
>
> To address the concern regarding the X-En translation direction, we have conducted additional experiments for the Fr-En pair. We invite you to refer to our general response to all reviewers, where the results are detailed in the accompanying table. We observe consistent improvements in the Fr-En direction with DiffNorm, which we hope will further convince reviewers of the effectiveness of our proposed strategy.
> We agree that extending our method to non-European languages would be intriguing, and our self-supervised approach should be readily applicable to monolingual Chinese or Japanese speech features. However, we are currently unaware of a suitable speech-to-speech translation (S2ST) dataset for English-Chinese or English-Japanese pairs. If you could point us to such datasets, we would be eager to include them in our future studies!
>
> ------
>
> References:
>
> [1] Lee et al., (2022). Textless speech-to-speech translation on real data.
>
> [2] Huang et al., (2023). TranSpeech: Speech-to-Speech Translation With Bilateral Perturbation
>
> [3] Ghazvininejad et al., (2019). Mask-Predict: Parallel Decoding of Conditional Masked Language Models
>
> [4] Gu et al., (2018). Non-Autoregressive Neural Machine Translation

---

> ### Comment · Reviewer_AuR5 · 2024-08-11
> **Thanks for the author response**
>
> I will keep my score.

---

### Official Review · Reviewer_g4Z7 · 2024-07-13

**Soundness:** 2
**Presentation:** 3
**Contribution:** 3
**Rating:** 6
**Confidence:** 4

**Summary:**

The authors introduce a process aiming to simplify the target distribution of speech-to-speech translation. This process uses a VAE model to map features to a latent space, followed by a diffusion model to normalize the features in the latent space. The authors use the generated dataset to train a non-autoregressive CMLM model on the CVSS-C dataset to validate its effectiveness.

**Strengths:**

1. The authors propose a novel speech normalization method using Denoising Diffusion Probabilistic Models.
2. The quality gain in En-Es direction is impressive.

**Weaknesses:**

1.  I am confused by the rationale behind using a diffusion model to normalize the speech representation. The authors first add noise to the speech representation (Forward Process) and then remove the noise (Backward Process). This design seems awkward to me. Why do you think this process of adding and then removing noise can help with normalization?
2. Using a Variational Autoencoder to map features to a latent space seems contradictory to the motivation of reducing data multimodality. The VAE provides an indefinite mapping, which may hinder efforts to reduce multimodality. Additionally, $z_0 = f(h; \theta_{enc})$ is not correct; the VAE provides a distribution over the latent space. A more suitable expression would be $z_0 \sim p(z; f(h; \theta_{enc}))$.
3. The experiments are only conducted on synthesised dataset. As the paper's contribution is to alleviate the multi-modality problem in data, conducting experiments on a real speech-to-speech dataset, like [1], is much better to support the major claims.
4. The baseline results (Conformer in EnEs) seem quite low compared to the reported results in the literatre.

[1] Lee et al., Textless speech-to-speech translation on real data.

**Questions:**

1. A simple CG method in En-Es can bring an improvement about 4.5 BLEU (Table 2, Line 7), any reasons behind it?
2. In Table 4, Acc-Rec, BL-Rec, and BL-Dn seem highly correlated when applying your method. BL-Dn performs well with minor perturbation and vice versa. However, when there is no perturbation, BL-Dn degrades dramatically. What is the reason for this? Is there a borderline value of $T$ that separates these two regions? I think this exploration is important for understanding the effectiveness of using the diffusion process to normalize speech.

**Limitations:**

The authors have addressed the limitations in Appendix E.

---

> ### Author Rebuttal · Authors · 2024-08-06
>
> Thank you for the time and effort you've invested in reviewing our paper. We are grateful for your recognition of our approach's novelty and the significant improvements it offers. Below are our responses to your comments:
>
> > Why do you think this process of adding and then removing noise can help with normalization?
> >
>
> By injecting noise and training models to denoise, Diffusion Models are capable of recovering noisy features. Due to the acoustic multimodality issue, where the same speech under varying acoustic conditions can exhibit slightly different speech features (refer to Figure 1(b) in [2] for examples), these models are particularly useful. Once well-trained, a Diffusion Model uses the prior information from the training data to reconstruct speech features. This is visually represented in Figure 5, where the more noise is injected, the more pronounced the "smooth" effect becomes. Thus, by corrupting the original audio and reconstructing it, we can mitigate its acoustic variation. Additionally, this process helps remove background noises when the original audio is noisy. Lastly, the self-supervised nature of diffusion-based normalization makes it widely applicable, as it only requires monolingual speech data and avoids the need for manual rules. This advantage was also acknowledged by reviewer AuR5.
>
> Beyond the normalization use case, the forward-backward process of the Diffusion Model has shown great potential in producing more robust and consistent features for other tasks as well, such as adversarial robustness [3], speech enhancement [6], and more.
>
> > The VAE provides an indefinite mapping, which may hinder efforts to reduce multimodality.
> >
>
> Thank you for the insightful discussion on VAEs! We agree that VAEs transform inputs into a probability distribution over a latent space which provides an indefinite mapping. However, such latent space is smoother and more regularized due to the Gaussian constraints. In the end, whether VAEs help reduce the multimodality depends on the nature of input data. We believe your point is valid that VAE does not guarantee the reduction of multimodality **given very complex data distribution.** However, in the context of speech translation, where prosody information is generally consistent, we utilize the regularized latent space to capture linguistic information and filter out unwanted details, thereby mitigating acoustic multimodality.
>
> Additionally, we are utilizing a combination of Variational Autoencoders (VAEs) and a Diffusion Model to achieve our objectives. The denoised features produced by the Diffusion Model are cleaner and more robust. Our empirical results have further verified the effectiveness of our DiffNorm strategy, demonstrating its capability to enhance the quality and consistency of the processed speech data.
>
> > A more suitable expression would be 𝑧0∼𝑝(𝑧;𝑓(ℎ;𝜃𝑒𝑛𝑐))
> >
>
> Thanks for pointing out the notation issue, we have corrected the formula in the draft.
>
> > conducting experiments on a real speech-to-speech dataset, like [1], is much better to support the major claims
> >
>
> Thank you for the suggestion. Due to resource constraints, we are unable to conduct further experiments on the larger-scale data used by Meta AI's researchers in [1]. However, we would like to highlight the following points:
>
> 1. We have benchmarked our method against the speech normalization method proposed in [1] using the same dataset (CVSS). Our results, as shown in Table 2, demonstrate that our method is more effective.
> 2. A significant portion of the data in [1] is mined S2ST data following [4], sourced from the same CommonCrawl project as the French and Spanish data used in our study. This similarity in data sources leads us to anticipate that our diffusion-based normalization will also be effective on the dataset used in [1].
> 3. Additionally, we have conducted extra experiments for the French-English (Fr-En) translation direction and have provided additional data points that further validate the effectiveness of our method. For detailed results on Fr-En, please refer to our general response to all reviewers.
>
> > The baseline results (Conformer in EnEs) seem quite low compared to the reported results in the literature
> >
>
> As indicated in the caption of Table 2, our baseline results for the Conformer-based model are sourced from the original TRANSPEECH paper (https://arxiv.org/pdf/2205.12523), specifically corresponding to the Basic Conformer Model (ID=3 in Table 1 of the TRANSPEECH paper). **Could you please provide clarification regarding the differing results you have observed in the literature?**

---

> ### Author Response · Authors · 2024-08-06
> **Rebuttal by Authors (Cont'd)**
>
> This comment follows from **Rebuttal by Authors**.
>
> > A simple CG method in En-Es can bring an improvement about 4.5 BLEU (Table 2, Line 7), any reasons behind it?
> >
>
> We are also intrigued by the significant improvement from CG on the En-Es dataset. We suspect this may be due to two factors: (1) the relatively small size of the En-Es dataset, which contains fewer than 80k data points for training, and (2) the longer sequence lengths of Spanish speech, averaging 256 tokens in the training set and 308 in the test set, which worsen the acoustic multimodality issue. CG operates by constraining the model to learn exclusively from the distribution of the target (Spanish) speech units, leading to a more consistent distribution for translation during inference. **This is particularly beneficial for longer target sequences.**
>
> To further support our claim, we recently extended our exploration of CG to English-German translation using the WMT14 dataset. Following the same settings as the CMLM model [5] and incorporating our proposed CG for the MT task, we observed that while the CMLM+CG model does not show significant improvement using the full WMT14 dataset, it does enhance performance by approximately 1 BLEU point when the training data is filtered to only include sequences longer than 60 tokens and the test data to only include sequences longer than 30 tokens, as demonstrated below:
>
> | Model | T=5  | T=10 | T=15  |
> |-------------------|------|------|-------|
> |          CMLM        | 19   | 20.2 | 20.5  |
> |               CMLM+CG    | 19.8 | 20.8 | 21.04 |
>
> For speech-to-speech translation that has much longer unit length, the effect of CG will be more noticeable, and therefore achieving large improvement over the baseline.
>
> > ….when there is no perturbation, BL-Dn degrades dramatically. What is the reason for this? Is there a borderline value of 𝑇 that separates these two regions?
> >
>
> We believe it is dataset-dependent to judge whether there is a borderline value of T that has drastic improvement. We performed more ablation studies using T=10 and T=30 as start time for noise injection and the result is shown below:
>
> | Start Time | Acc-Rec | BL-Dn |
> | --- | --- | --- |
> | T=10 | 93.8 | 15.98 |
> | T=30 | 91.6 | 17.29 |
>
> As indicated in the result table, with a very small T=10, the reconstruction accuracy is approximately 0.94, leading to a downstream ASR-BLEU score of 15.98. This represents a significant improvement compared to the baseline CMLM result of 12.58. The accuracy of 0.94 suggests that around 6% of tokens differ after reconstruction, which is likely due to the regularization effect of the VAE model, as T is too small for the diffusion model to cause reconstruction errors. This implies that the Spanish speech dataset may contain significant acoustic variations and noisy information, which the VAE model can already partially remove. With a larger T, the diffusion model further cleans up the representation, resulting in even higher downstream results (as observed from T=10 to T=100, where there is an increase in ASR-BLEU score).
>
> -----
> References:
>
> [1] Lee et al., (2022). Textless speech-to-speech translation on real data.
>
> [2] Huang et al., (2023). TranSpeech: Speech-to-Speech Translation With Bilateral Perturbation
>
> [3]Chen et al., (2024). Robust Classification via a Single Diffusion Model
>
> [4] Duquenn et al., (2021). Multimodal and Multilingual Embeddings for Large-Scale Speech Mining
>
> [5] Ghazvininejad et al., (2019). Mask-Predict: Parallel Decoding of Conditional Masked Language Models
>
> [6] Richter et al., (2023). Speech Enhancement and Dereverberation with Diffusion-based Generative Models

---

> > ### Author Response · Authors · 2024-08-12
> > **Invitation for Comments and Clarifications**
> >
> > Dear Reviewer g4Z7,
> >
> > We greatly value your feedback and have provided clarifiications and additional experiments and analysis. To ensure that we have properly addressed your concerns, we would greatly appreciate if you could review our responses and provide any further comments. We are looking forward to engaging with you before the discussion period ends.
> >
> > Thank you for your time and consideration.

---

### Author Response · Authors · 2024-08-06
**General Response to All Reviewers**

Thank you to all reviewers for their valuable feedback and for recognizing the innovation and effectiveness of our work. In response to reviewers AuR5 and DRdm, we conducted further experiments on the French-English translation direction to validate our method's effectiveness. The results are as follows:

| Model            | ASR-BLEU |
|------------------|----------|
| Transformer      | 15.44    |
| Norm Transformer | 15.81    |
| Conformer        | 18.02    |
| CMLM             | 16.95    |
| CMLM + BiP       | 18.03    |
| CMLM + DiffNorm  | 19.53    |

These results are consistent with our findings for En→Es and En→Fr, where the DiffNorm dataset showed a 2.58 improvement in ASR-BLEU score compared to the CMLM baseline. Due to limited computational resources and time, we have not yet conducted CG-related experiments, but we will include these results in the future. Additionally, we plan to release our code publicly, addressing suggestions from reviewers g4Z7 and DRdm, to facilitate its use with other languages and larger datasets.

---

### Decision · Program_Chairs · 2024-09-25

**Decision:**

Accept (poster)

**Comment:**

2 weak accepts, 1 borderline accept, 1 accept
strengths: interesting idea, wide applicability, solid improvements
weankesses: experiments can be extended to other tasks, novelty of paper can be improved.

still given the flaws i think this paper is wrong enough to accept.